# An Exploration of Pediatricians’ Professional Identities: A Q-Methodology Study

**DOI:** 10.3390/healthcare12020144

**Published:** 2024-01-08

**Authors:** Mao-Meng Tiao, Yu-Che Chang, Liang-Shiou Ou, Chi-Fa Hung, Madalitso Khwepeya

**Affiliations:** 1Department of Pediatrics, Chang Gung Memorial Hospital, Kaohsiung 83301, Taiwan; 2College of Medicine, Chang Gung University, Taoyuan 333, Taiwan; changyuche@cgmh.org.tw; 3Chang Gung Medical Education Research Centre, Chang Gung Memorial Hospital, Linkou, Taoyuan 333, Taiwan; a12031@cgmh.org.tw; 4Department of Emergency Medicine, Chang Gung Memorial Hospital, Linkou, Taoyuan 333, Taiwan; 5Department of Psychology, Chang Gung Memorial Hospital, Kaohsiung 83301, Taiwan; chifa@cgmh.org.tw

**Keywords:** pediatricians, professional, identity, Q-methodology

## Abstract

Professional identities may influence a wide range of attitudes, ethical standards, professional commitments and patient safety. This study aimed to explore the important elements that comprise pediatricians’ professional identities. A Q-methodology was used to identify the similarities and differences in professional identity. Forty pediatricians were recruited from two tertiary referral hospitals in Taiwan. A list of statements was developed by five attending physicians and three residents. R software was used to analyze the Q-sorts to load the viewpoints and formulate the viewpoint arrays. Additional qualitative data—one-to-one personal interviews—were analyzed. Twenty-eight of forty pediatricians, 11 males and 17 females, with an average age of 39.9 (27–62) years, were associated with four viewpoints. We labeled the four viewpoints identified for professional identity as (1) professional recognition, (2) patient communication, (3) empathy and (4) insight. The professional recognition viewpoint comprised of youngest participants—28–36 years—with the majority as residents (77.8%), while the empathy viewpoint comprised the oldest participants—38–62 years—with all as attending physicians. All participants in the empathy and insight viewpoints were married. This study found professional identity to be a multifaceted concept for pediatricians, especially in the areas of professional recognition, patient communication, empathy and insight into patient care.

## 1. Introduction

Physicians’ unprofessional behavior can affect patient–doctor relationships, patient safety and quality of care [1]. Such disruptive behaviors can further impact staff interactions, satisfaction and performance [2]. However, physicians identifying with themselves in their line of professional work is integral to personal improvement, better interaction with their colleagues and better patient-care outcomes [3,4]. Defined as a person’s viewpoint of who they are and serving as a basis for the application of special professional knowledge and skills, professional identity is crucial for achieving better patient outcomes. Professional identity comprises beliefs, values, motives and experiences used to define and guide our professional growth and skillful practices involving cognitive and moral reasoning [5,6]. Thus, medical education requires the acquisition of a professional identity as one of the necessary assets for novices [7]. The acquiring of relevant medical professional knowledge and skills is not only through university training but also through the attitudes, views and values developed during clinical practice [5].

The development of professionalism for pediatricians is very important. Pediatricians face high pressure from family members and the diversity and uncertainty that comes with caring for children who cannot express their concerns verbally, especially when ill [8]. Pediatric residents consistently immerse themselves in a high-stress work environment—3–5 years of training full of uncertainties and diversities [9]. Evidence has shown that pediatric care is associated with stress from disease changes and fatigue in dealing with family members’ anxieties [10]. Furthermore, studies have reported that residents who are depressed are more likely to make medical errors than their non-depressed peers [9].

Having a strong professional identity is a valuable asset that underpins positive coping strategies in residents during medical training [11]. Clinicians have experienced a variety of factors influencing their professional identity formation [12]. Professional identity formation is considered a fundamental process in the development of healthcare providers [13], and it is dynamic and not a fixed process [14,15]. This dynamism is key to practicing medicine, as the diverse experiences gathered contribute to identity formation, reinforcing one’s sense of self-esteem [14,16]. Therefore, building a resilient residency workforce that responds effectively to high work-related stress environments requires attention to the development of their professional identities [11].

However, professional identity is a continuous self-adjustment and regulating process that requires individuals to reflect in alignment with their identity standards. Self-regulation helps to improve psychomotor skills and is considered to reflect self-appraisal and self-reaction. This gives prominence to cognitively based motivators for professional identity [9]. The integration of professional identity formation into curricula positively influences professionalism [17]. To our knowledge, few studies have explored pediatricians’ professional identity formation [6,18,19]; however, none have fully conceptualized their identities.

Considering the understanding of a pediatrician’s professional identity is paramount to improving patient–doctor relationships, colleague interactions, work-related stress, self-regulation, and patient safety, the purpose of this study is to answer the question “what are pediatricians’ viewpoints around their professional identities?” using Q-methodology.

## 2. Materials and Methods

### 2.1. Study Design

Q-methodology was introduced in 1935 as a hybrid qualitative–quantitative research technique where participants provide meaning to statements through a sorting procedure [20,21,22]. This method overcomes some of the barriers to measuring attitudes and opinions [22]. Participants actively define themselves rather than being defined by a researcher’s prior knowledge or understanding. The viewpoint analysis of the data provides the means to account systematically for participants’ subjectivity and identify the characteristics of each belief system in the sample [22]. Q-methodology has been successfully used to answer key questions in medical education, including examining the professional identities of medical physicians [23,24,25].

### 2.2. Participants

The survey, conducted from August 2020 to July 2021, initially invited 60 pediatric attending physicians and 40 pediatric residents from two tertiary referral hospitals. Participants with a minimum of 1 year of work experience were included to provide statements about their personal feelings. In total, 40 participants completed the survey, comprising 24 attending physicians and 16 residents aged 27–62, with 25 females and 15 males. The study aims and procedures were explained to participants, including their rights to participate voluntarily. Written informed consents were obtained from the participants before their involvement. Privacy and confidentiality were ensured through the use of identification numbers. Ethical approval of the study was obtained from the Chang Gung Medical Foundation Institutional Review Board (No: 201902159B0C101). Following similar research in this area [23], study participants provided the following demographic information to help examine personal characteristics of the identified viewpoints: age, sex, marital status, non-clinical work, position rank in the hospital, and years of clinical practice.

### 2.3. Data Procedure and Collection

Definition of statements and Q-set development (Figure 1).

A list of data statements regarding pediatricians’ professional identity derived from the literature, media, conversations and purposive interviews was developed by five visiting staff and three residents from two tertiary hospitals. Each resident was from years 2, 4 and 5, respectively. The five attending physicians have experience ranging from 6 to 20 years, with individual durations of 6, 8, 10, 16 and 20 years. We held an explanation meeting on Q-methodology before their involvement in the study. The literature search was conducted using PubMed, while media, conversations and purposive interviews were carried out via e-mail and social media platforms such as “LINE”. The collection of statements continued until data saturation was reached, indicating no further emergence of new opinions. The search and conversations yielded a total of 125 statements: literature (48), media (29), conversations (17) and purposive interviews (31). These statements were further reviewed by the same panel of experts involved in the development. The experts engaged in a series of discussions, primarily through e-mail and social media, using an inductive (unstructured) approach with no pre-existing theory. The concourse achieved the desired outcome, and 54 statements (Q-set) that best represented the expression of pediatric professional identities were retained after removing overlapping statements and those not related to the topic. The 54 statements have been provided as a Appendix A.

#### 2.3.1. Validity

A pilot study was conducted to assess the wording and adequacy of the statements. The content validity index (CVI) was used to assess the clarity and relevance of the statements. Eight independent pediatrician experts in Taiwan rated the relevance of each statement on a 4-point Likert scale—1 = not relevant, 2 = of little relevance, 3 = relevant, and 4 = very relevant. An acceptable CVI rating was set at 0.8 or above [26]. Overlapping, misleading or ambiguous statements were revised or discarded, reducing the total from 125 to 54. The Q-set was conducted in Chinese language for participants to understand and engage in the process, and later, it was translated into English for writing purposes.

#### 2.3.2. Q-Sorting

A Q-methodology study was designed to extract the number of viewpoints through statistical analysis of a set of statements, identifying both distinctions and similarities between each viewpoint. This methodology was chosen to explore the link between those who prioritize specific viewpoints of professional identity, aiming for a broader understanding rather than merely categorization. The investigators first explained the Q-sort procedure, provided each participant with a list of Q-set statements along with explanations, and asked them to read before sorting. Participants were then instructed to familiarize themselves with the 54 Q-sort cards and distribute the statements on a Q-sort grid into 3 groups: least important, neutral and most important (Figure 2). The Q-sort grid is a visual table with patterns of distribution on a score sheet, with the number of spaces ranked and columns representing the spectrum from least important (1) to the most important (11) [20,23]. The design requires participants to carefully consider each statement to reflect its true meaning before placing it in each area of the score sheet. A Q-sort is finalized when all statements have been ranked on the Q-sort grid.

#### 2.3.3. Resorting and Post-Sort Interviews

After completing the Q-sort, a picture was taken to record arrays for each participant. Subsequently, we conducted individual interviews with participants to gather insights into the sorting process. Participants were asked a series of questions about why certain statements were considered most important or least important (Appendix A). Narrative commentary on the two statements ranked at each extreme of least or most important was provided to supplement interpretation and understand the reasons behind the statement ranking [24,27].

The entire process took approximately 30 min to 1 h, depending on how quickly the participants completed the sorting. After completion, the interviews were transcribed, anonymized, and linked to each participant’s unique Q-set for analysis. As a token of appreciation, participants received a USD 10 meal voucher. The collected Q-sorts were then subjected to a by-person viewpoint analysis, where similar Q-sorts were correlated into a unique viewpoint. By examining the defining statements within each viewpoint, the viewpoints were easily discerned. The Q-sorting cards and interview guides were initially prepared in Chinese and later translated into English for writing purposes.

### 2.4. Statistical Analysis

The qualitative data—individual interviews—were analyzed to illuminate the distinguishing statements and enhance the understanding of professional identities. The R software, specifically the “qmethod” package (ver. 3.6.1), was utilized to analyze the Q-sorts. This involved loading the statement scores for each participant, producing eigenvalues and formulating the viewpoint arrays (Appendix A). Descriptive statistics, including mean with standard deviation and frequency with percentage, were employed to summarize the demographics against each viewpoint.

### 2.5. Interpretation of Factors

The principal component analysis, readily available in R [28], along with varimax rotation, was employed for factor interpretation. Varimax rotation is a commonly used method that contributes to generating a clearer viewpoint and yields more distinguishing statements [27]. Viewpoints with an eigenvalue greater than 1.00 were chosen for optimal viewpoint identification [23,25,29]. The human judgment rule was also applied to determine the most appropriate number of viewpoints, if necessary [25,29]. In Q-methodology studies, a variability explained by the viewpoints of 40% or higher is considered acceptable [29]. The weighted average of the Q-sorts from the viewpoints indicated the statement on the Q-set grid. The distinguishing statements were uniquely important in reflecting a single viewpoint.

## 3. Results

Based on the eigenvalue and a thorough discussion with the team using the human judgment rule, we selected a 4-viewpoint solution to reflect participants’ perspectives on their professional identities. The viewpoint arrays were analyzed to indicate where each statement was placed on the grid and identify the distinguishing statements (Appendix A). This analysis helped us understand the hierarchy of importance for statements with each viewpoint. Out of the forty participants, twenty-eight (70%) were loaded onto the four viewpoints, while the remaining twelve (30%) were considered confounders and were not classified into any single viewpoint. The eigenvalues ranged from 3.8 to 8.4, satisfying the Kaiser–Guttman criterion—eigenvalue > 1 for significant viewpoint inclusion [29]. The four viewpoints explained 65.6% of the total variance, with viewpoints 1, 2, 3 and 4 explaining 21.1%, 20.3%, 14.8% and 9.4%, respectively (Table 1). The explained variance reported in this study surpassed the recommended 40% in Q-methodology studies [29].

The age of participants across the four viewpoints ranged from 27 to 62 years, with 17 females and 11 males. Detailed demographic information for each viewpoint is presented in Table 2.

We assigned names to each viewpoint as follows: viewpoint 1—professional recognition, viewpoint 2—patient communication, viewpoint 3—empathy, and viewpoint 4—insight. These names were based on the values and ideas prioritized in respondents’ explanations concerning their pediatric professional identities. The correlation among the four viewpoints was moderate to high, ranging from r = 0.38 to 0.61 (Table 3). Specifically, professional recognition had a lower correlation with empathy (r = 0.38) and insight (r = 0.41). Patient communication exhibited a high correlation with professional recognition (r = 0.61), empathy (r = 0.58) and insight (r = 0.59).

The viewpoint arrays demonstrated the placement of each Q-sort statement in terms of where they would be on the distribution grid for each viewpoint (Table 4). Each viewpoint is explained below, and excerpts from the interview data are provided where appropriate. Here, we also comment on the status of the rationales given and the extent to which they are presented as facts, opinions (e.g., hedged with phrases such as ‘I think’), or whether they are backed up with narrative accounts (i.e., stories of events experienced).

### 3.1. Viewpoint 1: Professional Recognition

The professional recognition viewpoint comprised 32% of the respondents, with two attending and seven resident physicians (77.8%). Their age ranged from 28 to 36 years, with 66.7% being married (Table 2). This viewpoint had the youngest participants as compared to other viewpoints. The two top statements identified for professional recognition included “*professional with keen observation ability*” (scoring +11) and “*can give family peace of mind*” (scoring +10), with the least prioritized statements that were distinguished as “*Learned knowledge and spiritual satisfaction from sisters*” and “*Centered on lifestyle and well-being*” (scoring +2), and “*Communication skills with the family*” (scoring +1) (Table 4).

The in-depth interviews revealed the top statements corresponded with participants’ own experiences and their medical knowledge for timely medical diagnosis (Table 5). The impact of what will be was also considered important for a logical medical diagnosis of diseases. In regard to the least important statements, this was based on personal opinions and facts—“I think”—which resulted in the statements being assumed to be true. Unexpected events, assigned teaching, and the ability to conduct research for professional recognition were not deemed clinically important, especially for residents.

### 3.2. Viewpoint 2: Patient Communication

The patient communication ability viewpoint comprised 32% of the respondents, with five attending and four resident physicians (44.4%). Their age ranged from 27 to 57 years, with most participants being females (77.8%) and about half (44.5%) married (Table 2). The two top statements for this viewpoint are “*soothe the child patients*” (scoring +11) and “*discuss with family as a partner*” (scoring +9), indicating a strong correlation between participants’ communication abilities and their application to clinical practice (Table 4). The least prioritized statements include “*communicate with peer for patients*” (scoring +5), “*centered on lifestyle and well-being*” (scoring +5), and “*extensive general knowledge*” not specific to pediatrics (scoring +3).

The in-depth interviews further clarified that the top priority statements were predominately based on participants’ opinions, although some of their rationales were backed up with narratives of their own experiences (Table 5). Thus, effective communication between doctors and patients contributes to the quality of patient care. Given that children cannot express and make their own decisions, obtaining consent and information from the parents becomes crucial. Regarding the least prioritized statements, communication with peers for patients’ related information, well-being lifestyle and extensive general knowledge was considered less important.

### 3.3. Viewpoint 3: Empathy

The empathy viewpoint, which focused on feelings, had participants who were the oldest compared to other viewpoints. Their age ranged from 38 to 62 years and were married (100%) (Table 2). This viewpoint comprised 22% of the respondents, with six attending physicians and no resident physicians. The two highest-ranked statements were “*doctors’ resilience*” (scoring +9) and “*working in a safe climate*” (scoring +8), while “*Pediatric patient-centered care*” was ranked the lowest with a score of +3 (Table 4).

During the in-depth interviews, the top statements were clarified as related to the family’s worry concerning the child’s disease and the family making all the decisions rather than the children themselves (Table 5). The least prioritized statements were related to children crying, which eventually interrupted the discussion between the doctor and family members, and the associated treatment care. The need for the doctor’s patience and empathy was emphasized.

### 3.4. Viewpoint 4: Insight

The age of participants in this viewpoint ranged from 34 to 52 years, and all were married (100%) (Table 2). This viewpoint highlights the inability of children to express their concerns and the varied reactions of family members to the disease. It comprised 14% of the responders, with three attending and one resident physician. The statements with the highest scores were “*ability to soothe children with good insight*” (scoring +11), “*Keen observation as a professional*” (scoring +9), and “*can know what the kids’ emotional response or wording is*” (scoring +8), while “*Things that kids care about*” scored with the least with +4, (Table 4).

Although the top statements were based on personal opinions and the family members being the key people in decision-making for the child’s treatment, the non-verbal cues from the child (e.g., face and extremity reactions) are vital for the pediatrician’s medical diagnosis (Table 5). This was further clarified during the in-depth interviews. Identifying child abuse is not easy from the family’s perspective but is achievable through the pediatrician’s keen observation. Discussing with peers to make a correct diagnosis based on patients’ insight is rather difficult and the primary reason for the least prioritized statements.

### 3.5. Consensus Statements

Overall, pediatricians’ professional identity is centered on concerns from the family and making the correct medical diagnosis of the disease. Considering the sensitivity of the child’s diagnosis and treatment and what information the family wants to know, effective patient communication abilities, empathy and insight are important for pediatricians’ professional identity and need to be cultivated further. Professional recognition is particularly crucial for younger pediatricians.

## 4. Discussion

Using Q-methodology, the present study aimed to understand how pediatricians’ professional identity is conceptualized. To our knowledge, this is the first study of its kind to fully explore such a concept using an extensive and rigorous process of developing a Q-set and validating it using factor analysis methods. Through the use of consensus and the steps for constructing a Q-set to explore pediatricians’ professional identity, this study provided deeper insight into how pediatricians can view their identities in the clinical context. We therefore identified four viewpoints that illuminate the distinctive ways pediatricians identify with their profession. In regards to the viewpoints, viewpoint 4—insight—indicated what the family is mostly concerned about; viewpoint 2—patient communication—highlighted physicians’ ability to effectively communicate the patient situation to the family members in a simple and clear manner; viewpoint 3—empathy—which emphasized the ability of physicians to share with others’ feelings and, thereby, imagining what it would be like to be in the family’s situation; and viewpoint 1—professional recognition—which reflected pediatricians’ clinical skill-set and knowledge, especially in the younger pediatricians’.

Apart from professional behavior, professionalism influences various levels of professional performance, including environment, competencies, beliefs, values, mission, and identity [1]. As defined, possessing a professional identity is an ongoing and dynamic process that continuously needs physicians to redefine their professional self-concept based on evolving attributes, beliefs, values and motives [7]. This is usually mediated by workplace and institutional discourses to understand the boundaries and hierarchies through an unfolding career and the unpredicted life stories [7]. Thus, gaining a better understanding of the nature, causes and impact of physicians’ behavior facilitates the development and implementation of appropriate supportive interventions for improvement [2]. Likewise, a better comprehension of pediatricians’ professional identities and identification of effective strategies to support these is required [1].

Professional recognition has been defined as the formal acknowledgment of an individual’s professional status and the right to practice in accordance with professional standards and subject to professional or regulatory controls [30]. Its system grew out of a desire and the need to recognize and reward professional nursing staff for their outstanding nursing care [31]. Evidence has shown the importance of positive recognition experiences in an organization, which needs to be fostered in order to buffer the negative effects of burnout [32]. In this study, having younger participants in the professional recognition than the empathy and insight viewpoints is a very interesting finding. Younger pediatricians consider the formal acknowledgment of professional standards status as the most important for their professional identities and are independent on different genders. Most of them are newcomers to the field, concentrating solely on their professional career trajectory and not prioritizing other factors such as empathy, communication, and insight.

A study by Luciano et al. [33] found the explanation should always come first in pediatric procedures. Most studies have reported explanations and communication to be more popular in nursing and child specialties [34]. Our study found that effective communication abilities are important for professional identities in some pediatricians, which is rare to be mentioned. Parents should, therefore, be involved in the decision-making process as a lack of effective communication often inhibits an open and mutual negotiation between families and their physicians [35]. The involvement is really important, especially when discussing difficult issues with patients and parents [36]. However, this study states that even in uncomplicated issues, pediatricians still need to have better explanation abilities to ensure children and their families understand what is going on.

As another important viewpoint in this study, empathy has been studied in various philosophical, psychological and social neuroscience studies as a complex interpersonal phenomenon [37]. Empathy involves caring, and its aspects include physicians acting as a bridge between them and the child, building a sheltered atmosphere, meeting the child’s needs, and adapting to the family’s life [38]. Providing expert physical care, fulfilling emotional needs and supporting daily parental care for the child is possible in a comfortable and inviting environment [38]. This is especially needed during resuscitative measures, where pediatric physicians have to provide the necessary psychosocial support [39]. During critical care situations, empathy is more important for better quality of patient care and its outcomes. Richardson et al. [40] argue that it is possible to teach students to use empathy when providing usual care in the clinical setting. This study found empathy is essential for pediatrician’s professional identities and for better disease explanation. This was further emphasized by the fact that the elders attending clinical services may have ample experience with patient needs and be mature enough to understand patients’ perspectives.

Having a comprehensive insight into the child’s care was also considered important for pediatricians’ professional identities. Decision-making concerning end-of-life care is not influenced by legal or economics but by insights from the family and patients to have better care [41]. The insights for pediatricians during medical diagnosis include using devices such as EEG to detect blood pressure abnormalities for loss of auto-regulation [42], gene study for autism spectrum disorders [43] and monitors to prevent cyberbullying [44]. In this study, non-verbal cues such as posture and emotional expression, including children’s language, were important insights for making an accurate medical diagnosis. A keen observation is very important as these cues can indicate a child and the family’s overtones.

### 4.1. Research as the Least Important

Our data found research is less important for clinical pediatricians PI, although >90% of them expressed having the research abilities. Similar to our finding, a study by Ullrich et al. [45] found that only 17% (14% versus 3%) of pediatric residents were likely to conduct clinical research and basic science or laboratory-based research. Surprisingly, our study also reported work–life balance, which is critical for individuals’ well-being, and lifestyle is not necessary for pediatricians’ professional identities. This is contrary to a study by Keeton et al. [46], which showed that career satisfaction can be attained without work–life balance, and this may be due to different years of practice [47].

### 4.2. Study Implications

Our study serves as a foundation for the development of initiatives designed to help pediatric physicians understand and foster their professional identities in the clinical setting. This, in turn, helps to create a community of practice that is coherent and achieves positive workplace outcomes, including patient care. The viewpoint of identities in this study further informs pediatric physicians on what areas to personalize and, possibly, allow the interchange of ideas among each other. This study found that according to age or years of experience, younger pediatricians emphasize clinical skill-set and knowledge, while elder attending pediatricians emphasize empathy for better disease explanation. Thus, the younger ones might learn empathy aside from focusing on skills acquisition for competency-based performance and vice-versa. Finally, this study contributes to professionalism literature by providing representative factors that influence professional identity. This creates a shared model and allows a comparison of professional identity across various specialties.

### 4.3. Limitations

This study was not without limitations. We only had two medical centers involved and a small number of participants, which limits the generalizability of results to other centers. A pre-existing theory was not used to fully construct the Q-set. However, a rigorous literature search and discussions were conducted to ensure a plausible and grounded framework for defining professional identity in this study. Despite such limitations, this study is the first study and provides the foundation for constructing a Q-set for professional identity to be used within the context of pediatrics.

## 5. Conclusions

This study highlighted the important concepts that define professional identity in pediatricians. Professional recognition, patient communication, empathy and insight are the most valuable concepts for professional identity in patient safety care. The younger pediatricians prioritize clinical skill-set and knowledge, whereas elder attending pediatricians emphasize empathy for better disease explanation, revealing a divergence in professional emphasis based on age and experience. The development of future education interventions across different pediatricians that might support key components of their professional identity needs to be further explored.

## Figures and Tables

**Figure 1 healthcare-12-00144-f001:**
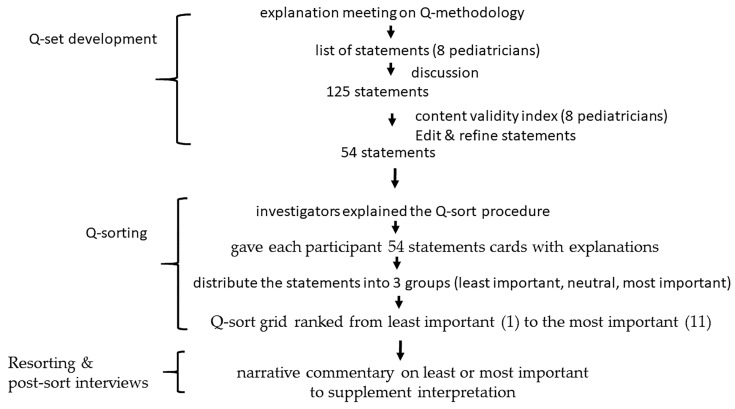
Q-methodology steps.

**Figure 2 healthcare-12-00144-f002:**
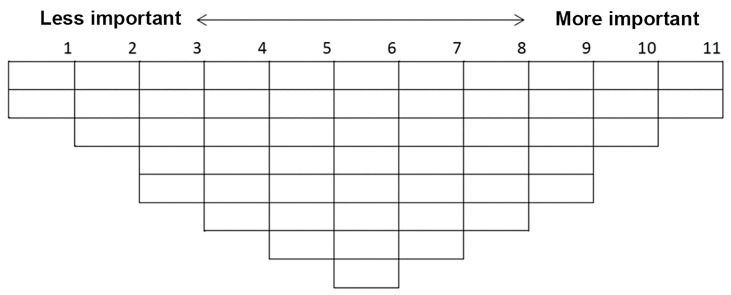
Q-sort score sheet for the 54 statements with a forced pattern of distribution.

**Table 1 healthcare-12-00144-t001:** Viewpoints identified for professional identity and their eigenvalues with 65.6% of the total variance.

Viewpoints (N)	Eigenvalue	Explained Variance (%)
1. Professional recognition (9)	8.4	21.1
2. Patient communication (9)	8.1	20.3
3. Empathy (6)	5.9	14.8
4. Insight (4)	3.8	9.4

**Table 2 healthcare-12-00144-t002:** Demographic details of the four viewpoints.

	Total	Professional Recognition	Patient Communication	Empathy	Insight
No. responder	28 (100%)	9 (32%)	9 (32%)	6 (22%)	4 (14%)
Age					
Range	27–62	28–36	27–57	38–62	34–52
Average	39.9	31.4	40.8	48.2	44.8
Years in Pediatrics (Average)	12.74	5.78	13.4	20.5	15.25
Sex					
Male	11 (39.3%)	4 (44.4%)	2 (22.2%)	3 (50%)	2 (50%)
Female	17 (60.7%)	5 (55.6%)	7 (77.8%)	3 (50%)	2 (50%)
Marital status					
Single	8 (28.6%)	3 (33.3%)	5 (55.5%)	0 (0%)	0 (0%)
Married	20 (71.4%)	6 (66.7%)	4 (44.5%)	6 (100%)	4 (100%)
Non-clinical work					
Teaching	19 (67.9%)	6 (66.7%)	6 (66.7%)	4 (66.7%)	3 (75%)
Research	26 (92.9%)	9 (100%)	8 (88.9%)	6 (100%)	3 (75%)
Rank					
Attending	16 (57.1%)	2 (22.2%)	5 (55.6%)	6 (100%)	3 (75%)
Resident	12 (42.9%)	7 (77.8%)	4 (44.4%)	0 (0%)	1 (25%)

**Table 3 healthcare-12-00144-t003:** Correlation of the identified professional identity viewpoints.

	Professional Recognition	Patient Communication	Empathy	Insight
1. Professional recognition	1			
2. Patient communication	0.61 ***	1		
3. Empathy	0.38 ***	0.58 ***	1	
4. Insight	0.41 ***	0.59 ***	0.60 ***	1

*** *p* < 0.001.

**Table 4 healthcare-12-00144-t004:** Q-set statements across each professional identity viewpoint, from highest to least ranked.

Viewpoints	Statements	Ranking
1. Professional recognition	(1) Keen observation as a professional	11
	(20) At the right time to give family a peace of mind	10
	(22) Discuss with family as a work partner	9
	(6) Pediatric patient-centered care	8
	(10) Things that kids care about	6
	(33) Can know what emotional response or wording is	3
	(38) Be patient with family members and sick children	3
	(49) Learned knowledge and spiritual satisfaction from sisters	2
	(51) Centered on lifestyle and well-being	2
	(48) Communication skills with the family	1
2. Patient communication	(16) The ability to soothe children	11
	(22) Discuss with family as a work partner	9
	(49) Learned knowledge and spiritual satisfaction from sisters	8
	(38) Be patient with family members and sick children	6
	(10) Things that children care about	6
	(31) Communicate with colleagues for patients	5
	(51) Centered on lifestyle and well-being	5
	(41) Extensive with general knowledge	3
3. Empathy	(34) Resilience	9
	(23) Safety climate for patient care	8
	(48) Communication skills with the family	5
	(16) The ability to soothe children	5
	(22) Discuss with family as a work partner	5
	(6) Pediatric patient-centered care	3
4. Insight	(16) The ability to soothe children	11
	(1) Keen observation as a professional	9
	(33) Can know what emotional response or wording is	8
	(6) Pediatric patient-centered care	8
	(23) Safety climate for patient care	8
	(38) Be patient with family members and sick children	6
	(51) Centered on lifestyle and well-being	5
	(48) Communication skills with the family	5
	(10) Things that kids care about	4

**Table 5 healthcare-12-00144-t005:** Excerpts of rationales provided by participants loaded into the professional identity viewpoints for high- and low-priority statements.

Viewpoints	High Priority	Low Priority
1. Professional recognition	F-R-29: [Medical knowledge] Since I am a pediatrician, I feel that even though I put that affinity, comforting ability, observation ability and other things to the back, I don’t think I can diagnose the disease. Hmm! Yes! So, I always felt more knowledgeable with things before, that is, at least the training process may be quite important to me, right. (resident physician)	F-A-42C: Unexpected assigned teaching; it is something that you will have as a doctor. It has nothing to do with the pediatrician, right. (resident physician)
	F-R-30C: [Quality of care] Then comes the abilities related to health and education diseases, because many family are worried as they don’t understand the disease, or if they happen to read some information on the Internet, they will see possible complications, etc. You’ll be more worried, yes, so it may be very important to educate the family on the disease whether it is now or later, on the impact or what will happen may also be quite important. (resident physician, CGU)	M-A-38: Research ability, in this matter, I think, this is not an ability that a pediatrician must have. (attending physician, CGU)F-R-30: Well, I think all of them are important at the moment, but not important to have the ability to do research, and it is difficult to have the credibility of research at the moment. Maybe this is very important to medical center physicians, but I think it is not clinically important at present. (resident physician)
2. Patient communication	F-A-49: The ability to communicate with doctors and patients, um, when some of our pediatricians communicate, he communicates from his own perspective completely, and the family does not understand it at all. Well, he didn’t care whether his family members could understand him or not, anyway, he just left after speaking like the anchor. Therefore, he actually didn’t care about the anxiety caused by his family members because he couldn’t understand it. That would not work. Well, I have seen this kind of doctor with my own eyes. (attending physician)	F-A-42: It is better to communicate with peer for patients’ related information, because there are other things that are necessary. I will rank other thing first if necessary. (attending physician)
	F-R-30: In addition, it is a broad and comprehensive knowledge and ability. The main reason is that although I think it may not really be a very specialization, it is because the sentence is comprehensive. It doesn’t necessarily require so much specialization, but in fact, it must be at each contact point. There is a way and most of them are to understand that most of the parents are like they are mastered. Hey, I think about how to explain it. It just doesn’t need to be very specialized, but it’s a little bit of contact for each and every field, that’s like it Is very extensive, but, at least the part that can communicate with family members may have a way to do one with them. Communication, at least we can know what they are talking about, and then we have a better way to pick out some language or something from it, there is a way to explain to the family, and to what extent th”y ca’ understand, hey. (resident physician)	
	F-A-36: Then discuss, yes! Because you will need a lot of time for children to discuss with their family, so I think this is very important. (attending physician)	
3. Empathy	M-A-36: Then comes the abilities related to health and education diseases, because many family are worried because they don’t understand the disease, or if they read some information on the Internet, they will see some possible complications. You’ll be more worried, yes, so it may be very important to educate the family on the disease whether it is now or later, the impact and what will happen may also be quite important. (attending physician)	F-A-49: Well, I don’t think I can ignore the cry of the kid! Yes! If he cries, children’s mother and I can’t continue, and the mother’s attention will be on him. So if there is a cry, we should make a decision, that is, whether to suspend first or what to do. (attending physician)
	M-A-56: It is because the family members who are engaged in pediatrics care with children patients, but in fact the real dominates is dependent on the family. The kids cannot have their own decision. (attending physician)	
4. Insight	F-A-44: It is because the family members who are engaged in pediatrics and to face their children patients, but in fact, the family are the real dominates. The pay attention to the patients’ expression even the face and extremities are more important. (attending physician)	F-A-49: The ability to discuss is important because what I know is still limited. I often say that I need to discuss the patient’s condition with my peers, so I can expand my limitations. Of course, the most important thing is to give the patient a correct diagnosis not only alert observation the patients. (attending physician)
	M-A-56: Because children don’t know how to talk! So sometimes the orders come from family members, but his general posture expression will be able to tell us the overtones. So you should have an extraordinary power of observation to be able to see what his overtones are. Because it is very important, they can’t speak, huh. (attending physician)	
	F-A-42: Because I think the pediatrician is less able to talk to the family, the facing of the children patients, so your observation skills are very important. (attending physician)	
	F-R-32: I think that keen observation is very important to the pediatrician, that is, you have to be like me in the previous cases of domestic violence, which is what you want to see, which is actually very important to him. (resident physician)	

## Data Availability

The datasets used during the current study are available from the corresponding author upon request.

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
