# Peer review of "An Exploration of Pediatricians’ Professional Identities: A Q-Methodology Study"

_healthcare, 2024, doi:10.3390/healthcare12020144_

Round 1

Reviewer 1 Report

Comments and Suggestions for Authors

The manuscript titled "Exploring Pediatricians' Professional Identities: A Q-methodology Study," submitted for consideration in Healthcare (ISSN 2227-9032), is well-structured, presenting a clear introduction, rationale, methodology, and discussion. The use of Q-methodology to explore subjective viewpoints is appropriate, and the findings are logically presented and discussed in relation to existing literature. The study makes a valuable contribution to the understanding of pediatricians' professional identity, although minor improvements in clarity and language could enhance its overall readability.

The manuscript exhibits commendable strengths in various key aspects. The objectives are exceptionally clear, aligning seamlessly with the research question and providing a well-defined roadmap for exploration. The literature review effectively highlights existing research gaps, emphasizing the necessity of the study, with the inclusion of stressors faced by pediatric residents adding depth and relevance to the research context. The methodology section demonstrates a meticulous approach to data collection, featuring a well-developed Q-set and a thorough validation process, including a pilot study and content validity index to enhance credibility. The statistical analysis is robust, employing appropriate methods with a clear justification for the chosen number of viewpoints. The results section is well-organized, presenting identified viewpoints and their characteristics, while the discussion effectively links findings to existing literature, emphasizing practical implications for pediatricians' education and professionalism. The conclusion provides a concise yet comprehensive recapitulation of key findings, highlighting the study's potential impact on pediatricians' understanding of professional identity.

However, I have identified a few concerns that require clarification:

  1. Why was one year of experience selected as an inclusion criterion for participants in the study?
  2. Can you provide details about the experience of the five visiting staff members on the expert panel?
  3. Did the expert panel members attend any workshops or courses to familiarize themselves with Q methodology before their involvement in the study?
  4. What specific roles did each of the five authors play in relation to this study, and how did they contribute to the research process?
  5. Were the eight independent pediatrician experts in Taiwan who validated the pilot study from the same tertiary hospital?
  6. The statement "Twenty-eight of the forty (70%) participants loaded on the four viewpoints, with the remaining 12 (30%) as confounders and therefore, were not classified into any single viewpoint" is not clear. Could you clarify what is meant by "confounders" and provide reasons why these 12 participants were not included in any single viewpoint?
  7. Considering the small sample size of 28 participants, it would be beneficial to explicitly mention this as a limitation in the study.
  8. Did the authors consider interpreting the Q methodology results separately for residents and attending physicians? If not, why?
  9. To enhance the readability of the results, consider using statistical diagrams or charts to explain the Q-methodology outcomes.

10.  While the in-depth interviews are mentioned, providing a few direct quotes or illustrative examples from participants could further enrich the qualitative aspects of the study.

In addition to these concerns, I suggest a final proofreading pass to address minor grammatical and typographical errors, and some sentences could be shortened for improved readability. The addition of visual aids, such as charts or graphs, could enhance the presentation of results and make complex information more accessible to readers.

In conclusion, I believe that addressing these concerns will significantly improve the manuscript's accessibility, clarity, and impact. I commend the authors for their hard work and the valuable contribution made through this research. I look forward to the revised version of the manuscript.

Comments on the Quality of English Language

The manuscript demonstrates a moderate proficiency in the English language. While generally clear and coherent, there is room for improvement in simplifying complex terminology for better reader comprehension. The sentence structure is adequate, but some could be refined for greater conciseness and clarity. A thorough proofreading is advised to address minor grammatical and typographical errors, ensuring a polished and professional presentation. 

Reviewer 2 Report

Comments and Suggestions for Authors

If we want analize safety and quality determinants in health care, especially in hospital wards and ambulatry care - understanding of doctors (for example pediatrician's) professional identity is paramount to improving patient-doctor relationships, colleagues’ interaction, work-related stress, 66 self-regulation and patient safety.

1. Introduction provides a background for analyzing the issue. In my opinion you should add more references about professional identity.

2. In section Materials and Methods; 2.2. Participants you should describe population of respondents, describe the number of physicians who completed the survey, and add two sentences on sociodemographic determinants.

3. 2.4. Statistical analysis is not clear for me - please explain.

4. Four tables, placed one below the other make the material unreadable, under each table there should be an analysis of the results.

5. Discussion is sufficient. I understand limitation of the study.

6. Conclusion is too short. You should add information and conclusion.

7. References need correction.

Reviewer 3 Report

Comments and Suggestions for Authors

The article exhibits strong writing skills and a clear presentation of ideas. However, there are areas where improvements could elevate its quality further. First and foremost, it would greatly benefit the readers if the authors included the tools or questionnaires utilized in data collection. This transparency not only enhances the credibility of the study but also allows others to replicate or build upon the research.

Additionally, shedding light on the translation process adopted would be immensely beneficial. Detailing the methodology and steps taken in translating the materials used in the study could offer insights into the rigor and reliability of the data collected.

Moreover, a crucial aspect that warrants attention is the justification behind the CVI (Content Validity Index) being 1, with no suggested alterations to items. Explaining why no modifications, rewording, paraphrasing, or deletions were deemed necessary can provide a deeper understanding of the meticulousness applied in the research process. 

The pursuit of perfection in academic work often stems from a dedication to accuracy and reliability. It's about ensuring that the findings and methodologies employed are robust, trustworthy, and capable of withstanding scrutiny. Providing such detailed justifications not only showcases the rigour of the study but also contributes to the academic discourse by encouraging transparency and thoroughness in research practices.

Reviewer 4 Report

Comments and Suggestions for Authors

Title: An Exploration of Pediatricians’ Professional Identities: A Q-Methodology Study.

Reviewer Comments: Authors aimed to explore the important elements that comprises pediatricians’ professional identities. A Q-methodology was used to identify the similarities and differences for professional identity among pediatricians. Forty physicians were recruited. Several statements were developed by fellow doctors. Four viewpoints identified for professional identity. 1) professional recognition 2) patient communication 3) empathy and 4) insight. In conclusion authors identified that professional identity as a multifaceted concept for pediatricians’ especially, in areas of professional recognition, patient communication, empathy and insight in patient care.

Strengths:

1.    Author did a good job in highlighting the professional identity in pediatricians.

2.    These kinds of studies assist in building a community that is coherent and achieves positive workplace outcomes.

3.    One of the advantages of using Q-methodology is that it gives the the opportunity to listen to various voices and fosters respect for the participants viewpoints regarding any topic.

Weaknesses:

1.    There is no molecular data or experimentally derived data.

2.    Sample size is not enough to come to the conclusions authors have claimed.

3.    These types of studies are prone to validity and reliability issues. 

4.    The questionnaire used in this study can be applied only locally or can be applied globally? 

5.    Did the pediatricians recruited in this study were working alongside behavioral health consultants (BHCs)? 

6.    The people who supported empathy have spent many years in the profession. Is it because they have enough experience with the patient needs and mature enough to understand parents’ perspective.  

7.    And the people who supported Professional recognition have spent few years in the profession. Is it because they don’t have enough experience with either patients or parents. Or since they are new to the field they are only focusing on their professional career trajectory and not focusing on other factors such as empathy, communication and insight? 

Round 2

Reviewer 3 Report

Comments and Suggestions for Authors

The authors have addressed the previous comments given on earlier version of the manuscript. I don't have any further comments.